# Prevalence and Modes of Transmission of Hepatitis C Virus Infection: A Historical Worldwide Review

**DOI:** 10.3390/v16071115

**Published:** 2024-07-11

**Authors:** Tommaso Stroffolini, Giacomo Stroffolini

**Affiliations:** 1Department of Tropical and Infectious Diseases, Policlinico Umberto I, 00161 Rome, Italy; tommaso.stroffolini@hotmail.it; 2Department of Infectious-Tropical Diseases and Microbiology, IRCCS Sacro Cuore Don Calabria Hospital, Via Don A. Sempreboni, 5, 37024 Negrar, Verona, Italy

**Keywords:** HCV, epidemiology, transmission

## Abstract

Hepatitis C virus infection affects over 58 million individuals and is responsible for 290,000 annual deaths. The infection spread in the past via blood transfusion and iatrogenic transmission due to the use of non-sterilized glass syringes mostly in developing countries (Cameroon, Central Africa Republic, Egypt) but even in Italy. High-income countries have achieved successful results in preventing certain modes of transmission, particularly in ensuring the safety of blood and blood products, and to a lesser extent, reducing iatrogenic exposure. Conversely, in low-income countries, unscreened blood transfusions and non-sterile injection practices continue to play major roles, highlighting the stark inequalities between these regions. Currently, injection drug use is a major worldwide risk factor, with a growing trend even in low- and middle-income countries (LMICs). Emerging high-risk groups include men who have sex with men (MSM), individuals exposed to tattoo practices, and newborns of HCV-infected pregnant women. The World Health Organization (WHO) has proposed direct-acting antiviral (DAA) therapy as a tool to eliminate infection by interrupting viral transmission from infected to susceptible individuals. However, the feasibility of this ambitious and overly optimistic program generates concern about the need for universal screening, diagnosis, linkage to care, and access to affordable DAA regimens. These goals are very hard to reach, especially in LMICs, due to the cost and availability of drugs, as well as the logistical complexities involved. Globally, only a small proportion of individuals infected with HCV have been tested, and an even smaller fraction of those have initiated DAA therapy. The absence of an effective vaccine is a major barrier to controlling HCV infection. Without a vaccine, the WHO project may remain merely an illusion.

## 1. Introduction

Hepatitis C virus (HCV) infection is believed to have originated between 500 to 2000 years ago, based on the analysis of molecular diversity [1]. Initially, it went undetected and was categorized within the non-A non-B (NANB) hepatitis group until 1989 when it was successfully isolated and cloned [2], marking it as the most recently identified pathogenic hepatotropic virus. In recognition of this significant achievement, Dr. Harvey J. Alter, Michael Houghton, and Charles M. Rice were awarded the Nobel Prize in 2020. HCV is an RNA virus classified within the family Flaviviridae.

The development of an immune assay for detecting circulating HCV antibodies [3] was instrumental in understanding the role of HCV and its impact on chronic liver disease [4]. Although the test described by Kuo et al. in 1989 [3] was not applied in clinical practice, it was pivotal in research to demonstrate the association between HCV positivity and post-transfusion hepatitis. Subsequent studies [5] led to the development of novel HCV antibody tests. Thanks to these researches, in the early 1990’s, an ELISA immunoassay (ELISA generation I) was commercialized with the intention of identifying individuals previously exposed to HCV. This test was later refined into generation II and III immunoassays with progressively increasing sensitivity and specificity. These tests were widely applied for blood product screening to ensure donation safety and for various serological research surveys that we describe in this article.

Over the past three decades, HCV has emerged as a major global health concern, with ongoing significance today. The World Health Organization (WHO) estimated that, in 2019, there were 58 million people worldwide living with chronic HCV infection, approximately 1.5 million (1.3–1.8 million) newly infected individuals, and 290,000 deaths attributable to HCV-related complications [6]. This figure reflected a decrease of 6.8 million viremic infections compared to the prevalence of 63.6 million in 2015, mostly due to the use of direct-acting antiviral agents (DAAs).

Significant progress has been made over the past three decades in understanding the prevalence, modes of transmission, diagnostic tools, and treatment of HCV infection. Currently, it has been estimated that, worldwide, only 21% of infected individuals have been diagnosed with HCV infection, and merely 13% of them have initiated DAA therapy [7], despite the high efficacy (over 95%) of this therapy in curing the virus, regardless of the stage of underlying liver disease or HIV coinfection [8].

Important changes have occurred over time in modes of HCV transmission. In the past, the infection mainly spread via blood transfusion and iatrogenic transmission due to the use of unsterilized glass syringes. Currently, in high-income countries, the risk of HCV transmission via blood transfusion is under control and iatrogenic exposure has been, to a major extent, reduced. Conversely, blood transfusion and non-sterile injection practices continue to play major roles in HCV spread in low-income countries, reflecting inequalities between different areas of the world.

Currently, injection drug use drives infection, particularly in high-income countries and areas where blood products are routinely tested for HCV [6,7]. Emerging high-risk groups are men who have sex with men (MSM), individuals exposed to tattoo practices, and newborns of HCV-infected pregnant women.

Recent review articles have focused their attention on various aspects of HCV infection [9,10,11,12]. Some have specifically addressed modes of HCV transmission, such as blood transfusion and other medical procedures [13], injection drug use [14], sexual transmission [15,16], and perinatal transmission [17].

This review exclusively concentrates on the evolving global pattern of HCV prevalence and transmission modes over time, highlighting their importance in defining preventive and curative measures, as well as identifying potential future vaccination strategies.

## 2. Prevalence

The accuracy and validity of overall and regional HCV prevalence figures may be compromised due to the predominantly asymptomatic nature of the infection, leading to underreporting unless actively screened. Additionally, in low- and middle-income countries (LMICs), the lack of surveillance systems in place due to cost and logistical constraints and the limited availability of laboratory assays for HCV testing do not allow for accurate detection of new and existing cases [6,7,9,10]. In particular, concerning the validity of data from the Africa region, there is uncertainty about its accuracy due to the varying surveillance systems across different African countries. In this context, factors such as access to testing, availability of infrastructure, and standardization of methodologies may influence the consistency and completeness of data collection.

HCV prevalence by WHO regions is shown in Table 1 [6].

Currently, 30 countries bear the brunt of the HCV burden, accounting for 80% of infected individuals, with the highest prevalence rates found in China, India, Pakistan, Ukraine, Russia, and the United States. Over 70% of viremic individuals reside in LMICs [7]. Significant disparities exist within WHO regions, such as in Europe, where central and eastern European countries like Ukraine and Romania have prevalence rates of 3.1% and 2.3%, respectively, whereas most Western European countries have rates below 1% [7,18].

The number of people undergoing treatment for chronic HCV has increased nearly ten-fold since 2010. Egypt alone accounted for nearly 3.6% of all treatments, corresponding to 3.5 million individuals, during the period of 2015–2020. However, Egypt’s global ranking for HCV infections decreased from fifth place in 2015 to seventeenth in 2020 [7]. Unfortunately, treatment access remains largely suboptimal in the majority of LMICs due to logistical and economic constraints.

### 2.1. Widespread HCV Infection in the Second Half of the 20th Century

The second half of the last century was characterized by widespread HCV infections due to iatrogenic transmission of the virus, particularly in LMICs (Table 2).

One of the most notable examples of widespread HCV infections occurred in villages across the Nile Delta (lower Egypt) during the 1950s to the 1970s, but it was only officially recognized in 2000 [19]. The overall prevalence of HCV was 24.3%, with rates increasing from 9.3% in individuals born after 1980 to over 60% in those born during the decade of 1941–1950. The widespread transmission of the infection within the population was linked to the extensive use of parenteral antischistosomal therapy (PAT) with tartar emetic [20].

Schistosomiasis has been a longstanding health issue in Egypt dating back several centuries [25], particularly affecting those residing along the Nile river. Dr. Frank provided a detailed account of the epidemic in *The Lancet* [20]. Intravenous tartar emetic emerged as the most commonly used PAT since the 1950s, especially during mass campaigns in the 1960s and 1970s. Unfortunately, during this period, injections were administered using reusable equipment that was inadequately sterilized, often boiled for less than 2 min. Between 1964 and 1982, more than 2 million PAT injections were administered annually to an average of 250,000 patients. These PAT campaigns, aimed at treating schistosomiasis, likely served as a vector for the spread of bloodborne infections such as HCV. Moreover, chronic schistosomiasis can compromise the immune system, hindering the clearance of HCV in infected individuals [26]. This period represents one of the largest examples of widespread iatrogenic transmission of bloodborne pathogens globally. Only in 1982 did oral praziquantel, a highly effective drug for treating schistosomiasis, become available in Egypt, gradually replacing PAT as the standard treatment for schistosomal infections by the late 1980s. The prevalence of HCV by year was nearly seven times higher in individuals born before the 1960s compared to those born after the 1980s, likely reflecting the replacement of PAT with oral drugs since the late 1980s. A survey conducted in the general Egyptian population in 2015 revealed lower prevalence figures: 10% for anti-HCV and 7% for HCV RNA positivity, attributed to mortality among older age groups with the highest infection rates [27]. More recent data (2018–2019) showed even lower prevalence rates, with 4.6% positive for HCV antibodies and 3.5% viremic individuals, reflecting widespread treatment with DAAs since 2015. However, these findings may not be representative of the entire Egyptian population as the study excluded individuals over 60 years old and the majority of participants were younger than 45 years [28].

Cameroon provides another example of iatrogenic HCV transmission due to the use of unsafe medical equipment for intravenous drug treatment. A study conducted in 1991 on the prevalence of anti-HCV in an urban child population (aged 4–14) showed an overall rate of 14.5%, decreasing from 17.5% in children born between 1977 and 1979 to 6.5% in those born between 1985 and 1987 (Figure 1).

In that study, specific modes of transmission were not systematically investigated [29]. Nearly two decades later (in 2009), a study conducted in southern Cameroon focusing on individuals aged 60 to 102 revealed an overall HCV prevalence of 56%, decreasing from 68.3% in subjects born before 1935 to 40.3% in those born between 1945 and 1949 (Figure 1) [21]. This widespread infection is believed to have originated around 1920 and was found to be associated (odds ratio = 2.2; 95% confidence interval = 1.1–3.4) with the historical use of intravenous quinine for malaria treatment. In the rainy and forested regions of southern Cameroon, the incidence of infective malaria vector bites is among the highest globally [30]. Intravenous quinine has been available since the early 20th century and was often administered intravenously rather than intramuscularly to prevent abscesses at injection sites [31], using inadequately sterilized medical equipment. The same author documented a significant spread of HCV (prevalence of 10.5%) in individuals born before 1951, even in the Central African Republic, attributed to the intramuscular administration of pentamidine for treating trypanosomiasis prior to 1951 [22]. During this period, disposable syringes were not available, as the risk associated with the transmission of infection through human blood was not fully understood. Reflecting on HCV dissemination, Dr. Thomas Strickland, a leading expert on infectious diseases in Africa, commented [32]: “The Egyptian Ministry of Health and Population had the best intentions when it launched mass treatment campaigns with intravenous tartar emetic to control what they considered to be the country’s most important disease, schistosomiasis. Likewise, French colonial doctors in the CAR and Cameroon also had the best of intentions 50–80 years ago when they treated patients for endemic infectious diseases, including frequently fatal sleeping sickness and malaria; good intentions that led to serious, tough unintended, global health consequences”.

The spread of HCV infection through the intravenous treatment of leprosy in the past, utilizing unsafe and reusable syringes, has been documented in Africa [33], Yemen [33], and Brazil [34]. Similarly, in some high-income countries, HCV transmission occurred in the past due to the use of unsafe medical equipment. For instance, in a remote area in Japan, a remarkably high rate of HCV was reported in 1994 [24]. The overall prevalence was 32.4%, with rates increasing from 0 in subjects aged 0–15 years to 44.9% in those older than 40 years (Table 3). This transmission was notably associated with practices such as acupuncture and non-sterilized knives used for cutting.

In a town in southern Italy, the overall prevalence of anti-HCV antibodies in 1996 was 12.6%, with rates decreasing from 33.1% in individuals born before 1937 to 1.3% in those born after 1966 (Figure 2) [24].

The high rate in the oldest age group reflects a cohort effect, i.e., decreased risk of infection along generations as a consequence of improved sanitary conditions. The use of non-disposable syringes was common for medical treatment until the early 1970s. Subjects reporting this exposure were two-fold (CI 95% 1.03–3.4) more likely to be HCV-positive than those who were not exposed. Avoidance of non-disposable syringes over the last decades has generated a lower HCV prevalence in the youngest generations, as observed in a survey reported 14 years later in the same town [35] (Table 4).

### 2.2. Prevalence by Age

The age-specific prevalence of HCV is significantly influenced by the predominant modes of transmission at any given time. Currently, HCV primarily affects older generations in lower-, middle-, and to a lesser extent, high-income countries. By contrast, the peak prevalence among young adults reflects the importance of intravenous drug use in the transmission of the infection. This is particularly evident in the United States, where during the period of 2003–2010, the estimated prevalence of HCV RNA-positive individuals over 20 years of age was 1% (equivalent to 2.68 million people) [36]. Among them, 1.09 million individuals (40.7%) were aged 40–49 years. HCV RNA-positive individuals aged 20–59 years were 8.7 times more likely (95% CI 5.9–12.8) to report injection drug use [36].

The changing pattern of modes of transmission over time within the same country also influences the age-specific prevalence of HCV. For example, in Italy, a study conducted in five metropolitan areas revealed two peaks of infection prevalence: a higher peak (7%) among individuals born between 1935 and 1944 (representing the tail of the cohort infected in the past by unsafe medical equipment) and a smaller peak (1.6%) among those born between 1965 and 1974 (representing the cohort of injection drug users) [37] (Figure 3). In this survey, individuals reporting a history of intravenous drug use were 30.2 times more likely to be HCV-positive (95% CI 12.7–71.9) [37].

### 2.3. Prevalence by Sex

HCV infection is more prevalent in males, with a male-to-female ratio of 2:1 [38,39].

A potential explanation of this phenomenon may be the higher prevalence of males as compared to females among subjects reporting injection drug use.

Male gender is also associated with more severe progression of the disease [40]. Studies have shown a protective effect of estrogen on fibrogenesis, attributed to the inhibition of stellate cell proliferation [41,42]. In Egypt, clearance of HCV (defined as positive for HCV antibodies and negative HCV RNA test results) was found to be 1.8 times more likely to occur in women (44%) than in men (33.7%) [43]. However, it is important to note that these findings were derived from a cross-sectional study, which may be influenced by survival bias [44].

## 3. Modes of Transmissions

The major modes of transmission for HCV include unscreened blood and blood products, injection drug use, non-sterile injections, and unsafe medical procedures in healthcare settings. Additionally, to a lesser extent, transmission can occur through sexual intercourse, vertical transmission (from mother to child), and certain beauty treatments. It is important to note that the significance of these modes of transmission can vary over time and across different health settings (Table 5).

### 3.1. Blood and Blood-Derived Transfusion

During and after World War II, blood transfusions were widely utilized, followed shortly thereafter by the administration of blood derivatives [45]. The so-called non-A, non-B (NANB) hepatitis was found to be responsible for up to 90% of cases of transfusion-associated hepatitis [46]. Until the late 1980s, blood and blood-derived products were not screened for HCV, resulting in a transmission rate of approximately one case in every 50 blood units, even in high-income countries [47]. Shortly after the introduction of the first assay for HCV detection (first-generation ELISA) in blood donors, subsequent generations of assays (second- and third-generation ELISA) nearly eliminated the risk of HCV transmission through blood transfusion [48]. These newer assays were highly sensitive, reducing the serological window phase of detection. Indeed, the residual risk of HCV transmission is closely associated with units collected during the donor’s serological window period; the shorter the window period, the lower the residual risk [13].

The availability of serological assays led to the identification, several years later, of two HCV epidemics that occurred in pregnant women during the second half of the 1970s. Both epidemics were linked to the intravenous administration of batches of anti-HD immunoglobulins contaminated with HCV in Ireland [49] and Germany [50]. These two epidemics also provided valuable information on the natural history of HCV infection in female patients. After follow-ups of 17 and 25 years, respectively, only 2% of the Irish cohort and 0.5% of the German cohort had developed liver cirrhosis [50,51].

The introduction of nucleic acid testing (NAT) during the 2000s has significantly enhanced the safety of blood transfusions, reducing the risk of acquiring blood-transmitted viruses such as HIV, HCV, and HBV. In the pre-NAT era, the residual risks per one million donations were 16.7 for HCV, 1.9 for HIV, and 69.2 for HBV [51]. However, in the period between 2009 and 2015, the residual risks were dramatically reduced to extremely low levels, estimated to be 1 in 12,979,949 donations for HCV, 1 in 1,917,250 donations for HIV, and 1 in 2,555,854 donations for HBV [52,53,54] (Table 6).

Indeed, blood supplies are highly safe in high-income countries. Conversely, the situation remains concerning in low- and middle-income countries, where this mode of transmission continues to be a significant issue and no major database derived from systematic surveillance is available. The situation in Sub-Saharan Africa (SSA) presents a formidable challenge, as it combines endemic bloodborne infections, scarce resources, and a high demand for transfusions. Notably, up to 8.7% of blood units are discarded due to reasons related to transfusion-transmitted infections (TTIs) in the WHO African region [55]. Furthermore, data regarding TTI incidence and prevalence are often limited, reflecting local contexts, with estimates of transfusion transmission rates either absent or outdated. One of the primary hurdles lies in operational resources as the consistent implementation and maintenance of systematic serological and molecular blood screening necessitates a steady supply of test kits, reagents, and consumables, contingent upon reliable procurement and supply systems. However, transfusion services frequently lack control over procurement, leading to fluctuations in suppliers and irregular supply chains. Reports of sporadic supply and shortages of reagents in blood centers are widespread across SSA, particularly in rural areas [55]. On top of this, blood units are often not adequately screened, and in some cases, they are provided by paid blood donors. Additionally, there is a substantial demand for transfusions, especially in rural and forested areas of Sub-Saharan Africa, due to the co-existence and co-evolution of malaria and sickle cell disease. The entire process of blood testing remains unsafe in these regions [56] and the implementation of NAT screening is not feasible due to cost inefficiencies. The high cost and limited availability of this screening tool pose significant barriers to its adoption in low- and middle-income countries [57].

### 3.2. Injection Drug Use

The availability of commercial tests for HCV detection since the early 1990s has effectively reduced HCV transmission through blood transfusions. Consequently, injection drug use has become the major route of HCV transmission, especially in high-income countries, and it is emerging even in rural areas [58]. The sharing of equipment used for injecting drugs is the primary mode of virus transmission in this context. Worldwide, it has been estimated that there are 15.6 million people who inject drugs (PWID), with a predominance of males (12.5 million) compared to females (3.2 million). Among these individuals, 52.3% are infected with HCV, while 17.8% are infected with HIV. The highest HCV prevalence is found in Eastern Europe (64.7%), while the lowest is in Sub-Saharan Africa (21.8%) [59].

From 1992 to 2021, the pooled HCV incidence among PWID was reported at 12.1 per 100 person-years (which is 7.1 times lower) [14].

HCV infection is more likely to spread in high-income countries than in LMICs, although the population of PWID is expanding in Sub-Saharan countries [14]. If the risk of HCV transmission among PWID was removed, it is estimated that approximately 79% of cases in high-income countries and 38% of cases in LMICs would have been prevented during the period of 2018–2020 [60]. This reflects the differing proportions of PWID according to the income level of countries.

The likely higher prevalence and incidence rates of HCV compared to HIV among PWID require further comment. A plausible explanation was provided in 2014 [61]: HCV remains infectious at room temperature for up to six weeks compared to HIV, and HCV can survive longer in high-volume syringes. Consequently, the risk of transmission through contaminated needles and syringes after only needle exchange may persist for a longer period with HCV compared to other viruses. Moreover, the amount of HCV virus able to transmit the infection through contaminated equipment is much smaller than that of HIV, which also contributes to increased efficiency of HCV transmission among injection drug users.

Needle and syringe exchange programs (NSPs) play a crucial role in reducing HCV transmission in high-income countries. The prevalence of HCV among PWID younger than 30 years of age decreased from 44% in 2004 to 9% in 2011 [62]. Unfortunately, similar programs are rarely implemented in Africa, where only 1.1% of countries have comparable needle and syringe exchange programs [63]. The last available data on the topic generally affirmed the effectiveness of NSPs in reducing HIV and HCV infections in LMICs. Moreover, when high coverage is attained, NSPs seem to be as efficacious in LMICs as they are in high-income countries [64].

Considering these factors, implementing interventions to enhance HCV testing, facilitate linkage to care, and initiate treatment among people who inject drugs presents significant challenges. A recent systematic review and meta-analysis evaluated interventions aimed at improving outcomes such as testing, linkage to care, and treatment initiation [65]. Patient education and navigation were effective in addressing patient-level barriers to HCV care by improving antibody testing uptake and linkage to care, respectively. Provider care coordination proved effective in improving antibody testing uptake. Additionally, three different interventions targeting system-level barriers showed effectiveness at various stages of HCV care: point-of-care antibody testing (for linkage to care), dried blood-spot testing (for antibody testing uptake), and integrated care (for linkage to care and treatment initiation) [65]. Coupling these interventions with the above-discussed factors should be considered when designing actions toward HCV elimination. The implementation and effectiveness of preventive measures, such as harm reduction programs for intravenous drug users or the promotion of safe medical practices, may benefit from this analysis. On the other hand, among the factors reducing the efficacy of these programs, we should not leave behind potential armed conflicts in areas of high endemicity for HCV and forcible displacement of injection drug users that may benefit from opioid agonist therapies. Recent data have shown how these services suffer from political instability, leading to low retention in care and potentially high HCV-related morbidity and mortality [66].

### 3.3. Iatrogenic Transmission

The use of non-sterile medical equipment has historically led to significant HCV epidemics in both LMICs and high-income countries, as described in previous paragraphs detailing epidemics and TTIs. Still, the healthcare environment continues to pose a risk, even in high-income countries. Recently, an increased risk of HCV transmission following exposure to invasive procedures was documented [67]. Endoscopy, in particular, has been associated with a nearly four-fold increased risk of HCV acquisition. With these procedures being performed more frequently, a significant number of individuals may acquire infection through this route. Inadequate sterilization of instruments likely facilitates patient-to-patient transmission. The potential persistence of viremic HCV in the environment for more than six weeks [61] underscores the need for strict adherence to safety procedures.

Conversely, adherence to standard safety procedures was shown to prevent HCV transmission among 912 subjects undergoing endoscopy with properly disinfected instruments, despite their use also in HCV-positive patients [68]. In LMICs, the reuse of injective equipment and medical devices remains common. The WHO estimates that nine billion injections are given with reused equipment [69], largely due to financial constraints [70]. Additionally, in several African countries, the Middle East, and South Asia, the ongoing practice of genital female circumcision with unsafe instruments plays a central role in HCV transmission [71]. Increased surveillance should be instituted to monitor these unsafe practices and to identify new potential techniques that may lead to infection.

### 3.4. Sexual Transmission

The efficiency of HCV transmission can vary depending on the type of sexual intercourse. Among monogamous HIV-negative couples, transmission from an HCV-positive partner to an HCV-negative partner is rare, considering discordant genotypes in couples with both partners infected [15]. Studies suggest that HCV transmission among spouses may be more likely due to common exposure to percutaneous risk factors, such as sharing personal hygiene items and glass syringes, rather than through sexual intercourse. Couples where both partners are HCV-infected were 12.4 times more likely to report sharing these items compared to couples with only one infected partner [72].

A prospective Italian study involving 767 negative partners of HCV-positive individuals followed up for 10 years showed a low incidence rate of infection, as low as 0.37 per 1000 person-years [73]. All participants in this study were advised not to share personal hygiene items such as toothbrushes, nail clippers, and razors. Another American study quantified the risk of HCV transmission among monogamous couples, finding a low rate of viral transmission between spouses, with a rate of transmission of 0.07% per year or 1 per 190,000 sexual intercourses, without specific sexual practices associated with HCV transmission [74]. These findings provide reassuring counseling messages for monogamous couples, particularly when no sexual intercourse occurs during concomitant viral or bacterial infections [15].

However, the risk of sexually transmitted HCV is higher in cases of heterosexual condomless activity with multiple sexual partners [75]. An Italian study found that individuals reporting three or more sexual partners in the previous six months had a nearly three-fold higher risk of acute HCV compared to those reporting only one sexual partner, after adjusting for potential confounders such as other exposures and excluding subjects reporting blood transfusion and intravenous drug use [76]. Coinfection with sexually transmitted bacterial diseases (STDs), which can disrupt genital mucosa integrity, further increases the risk of HCV transmission [15].

Men who have sex with men (MSM) are at the highest risk of sexual transmission, especially if coinfected with HIV [77]. In this context, traumatic damage to the rectal mucosa facilitates viral transmission [15]. A recent systematic review quantified the prevalence and incidence of HCV in MSM according to HIV status [77]. The overall HCV prevalence was 3.4%, with the highest prevalence rates in Africa (5.8%) and Southeast Asia (5%). Stratifying by HIV status, the prevalence ratio was 4.2, with prevalence rates of 1.55% in HIV-negative MSM and 6.3% in HIV-positive subjects. In HIV-negative MSM, the HCV incidence rates were 0.12 per 1000 person-years without pre-exposure prophylaxis (PrEP) and 14.8 per 1000 person-years in subjects on PrEP. The HCV incidence in HIV-positive MSM was 8.46 per 1000 person-years. Indeed, the burden of HCV varies by region according to the prevalence of PWID and HIV. Moreover, using data from a prospective observational cohort study involving HIV-positive MSM with acute HCV infection, the incidence of HCV reinfection after spontaneous clearance (SC) or successful treatment was evaluated, revealing that a high incidence of HCV reinfection among HIV-positive MSM was strongly linked to sexual risk behavior, underscoring the urgency for interventions to curb risky behavior and prevent reinfections. Specifically, the reinfection rate stood at 11.5 per 100 person-years, with a median reinfection time of 1.3 years (IQR 0.6–2.7). Factors associated with HCV reinfection included receptive condomless anal intercourse, sharing of sex toys, engaging in group sex, anal rinsing before sex, having ≥ 10 casual sex partners in the last six months, and a nadir CD4 cell count < 200 cells/mm^3^ [78]. To dig deeper in this special population, we should consider data from another very recent observational European cohort that assessed SC in a specific sub-population of HIV-positive MSM with recently acquired HCV infection (RAHCV). SC of RAHCV in HIV-positive MSM was found in only 11.9% of cases and immediate DAA treatment for RAHCV infection may be favored in patients with ongoing transmission risk behaviors such as those described above [79]. All of these factors should be considered when designing public health campaigns and individual assessment on the road to HCV elimination programs.

Figure 4 illustrates the gradient of increasing risk of sexual HCV transmission according to the setting of intercourse. In all settings, the use of condoms is advisable, except for monogamous couples without other risk factors such as PWID, HIV, or STDs.

### 3.5. Perinatal Transmission

As per data derived from US surveillance, around 29,000 HCV-positive women give birth in the USA annually, but global figures for HCV in pregnant women are lacking [80]. In 2019, AASLD–IDSA guidelines, subsequently updated in 2023, recommended routine HCV testing for pregnant women, proving cost-effective if HCV prevalence among them is 0.03% or higher [81]. This aligns with the 2020 universal screening recommendations by the US Preventive Services Task Force [82]. Currently, obstetrical societies and general screening recommendations, including the WHO, mostly rely on targeted risk or birth cohort-based strategies to detect HCV in pregnant women, but screening practices are expected to change as countries work toward HCV elimination goals. In the document “Global Hepatitis Report 2024: Action for Access in Low- and Middle-Income Countries,” it is noted that most reporting countries have adopted and are implementing a national hepatitis C testing strategy or policy in line with WHO guidelines. While nearly all (90%) provide targeted testing for individuals from the most severely affected populations, only six countries focus their efforts on testing pregnant women. Consequently, implementing these specific testing strategies remains challenging in low- and middle-income countries [83]. Furthermore, risk factors for HCV infection, such as female genital mutilation, prevalent in various African regions, may not be adequately accounted for in existing methodologies [84]. Moreover, it has to be considered that only anti-HCV-positive women with viremia face the risk of mother-to-child transmission (MTCT) of HCV, which is further heightened by HIV coinfection [84]. The degree of HCV viremia correlates with MTCT risk, though a precise threshold remains unidentified, fueling uncertainty in the best strategy to adopt. MTCT can occur during labor, around childbirth, or postpartum [82]. In particular, the risk of mother-to-infant transmission is six-fold higher in HCV monoinfected mothers and doubles in HCV-HIV coinfected mothers [85]. Transmission primarily occurs during delivery, particularly in the presence of high maternal viral load during the third trimester of pregnancy [9,84]. Additional factors such as amniocentesis, prolonged rupture of membranes, and invasive fetal monitoring may also contribute to transmission [17,86]. The mode of delivery and type of feeding do not significantly affect vertical transmission in monoinfected women [11].

Direct-acting antiviral (DAA) therapy, with a sustained virologic response (SVR) rate exceeding 95% [8], has the potential to prevent vertical HCV transmission, provided universal HCV screening in pregnant women [87]. However, testing and linkage to care remain inadequate [88]. Sofosbuvir-based therapies initiated during the second trimester have been found to be safe and well-tolerated in real-life cohorts [88] and in a phase 1 study [89].

### 3.6. Body Grooming and Modification

Tattooing and body piercing are prevalent practices, especially in high-income countries, spanning across various social classes and age groups. These practices pose a risk for transmitting various infectious diseases, particularly bloodborne viruses such as HCV. Contamination of instruments with infectious blood can occur, and if proper sterilization is lacking, this can lead to the transmission of infections to other individuals. Studies conducted in different countries [90,91,92,93,94] have consistently demonstrated a significant association between tattooing and HCV infection, with closely aligned levels of risk (Table 7).

Tattooing plays an important role in HCV transmission particularly in the prison setting. It is well known that prison inmates receive tattoos while incarcerated as a sign of gang affiliation or other personal reasons, often using non-sterilized or rudimental instruments, increasing the risk of HCV transmission in a setting where the virus is already prevalent [95].

An Italian study [94] was unique in its recruitment, as it surveyed acute HCV cases and also investigated the role of piercing, revealing a significant association (odds ratio: 2.4; 95% confidence interval: 1.2–4.8) with HCV. These routine procedures, frequently conducted using unsafe methods, may contribute to a significant proportion of HCV cases. Therefore, the use of disposable instruments and proper sterilization of equipment are imperative measures to mitigate this risk.

In Pakistan, shaving with the same blade has been shown to be a strong independent risk factor (O.R. 5.1; 95% C.I. = 1.5–17.0) for HCV transmission among males [96].

Unfortunately, recent data in high-prevalence settings demonstrated that overall awareness among workers in beauty salons about parenterally transmitted viruses, including hepatitis C, is inadequate [97].

### 3.7. Attempts to Eliminate HCV Infection

Primary prevention interventions for HCV are of utmost importance [98]. Implementing screening of blood supplies [99], ensuring safe injections in LMICs [100], implementing harm reduction interventions such as needle and syringe programs for PWID [101], and employing behavioral interventions for MSM [102], are all measures that have the potential to impact the likelihood of HCV transmission. As discussed above, harm reduction initiatives for intravenous drug users and advocacy for safe medical practices provide a robust foundation for curtailing HCV spread. Understanding the implementation nuances and efficacy of these preventive measures are imperative for a holistic approach to disease prevention [65]. To anticipate and adapt to emerging trends and challenges in the realm of HCV transmission and prevention, the evolving patterns of drug use, advancements in treatment modalities, and the integration of digital health technologies should not be neglected in the fight for HCV prevention and control. Innovative approaches and sustained collaborative efforts across sectors are being implemented at different levels. Recent experiences integrated traditional with machine learning-guided syringe service programs in order to identify high-priority areas for implementation due to their increased vulnerability, in both urban and rural communities outside of current existing programs [103]. Moreover, accurate spatiotemporal mapping of drug overdose deaths was achieved through machine learning analysis of drug-related web searches, integrating existing programs for injection drug users in the USA [104]. Additionally, in Europe, neural network analysis of socio-medical data or questionnaires was employed to identify predictors of undiagnosed hepatitis C virus infections and develop potentially more efficient risk-adapted HCV screening [105,106]. These techniques have the potential to enhance resource allocation and program prioritization, reshaping national initiatives in both high- and low/middle-income countries. However, the debate regarding the use of such technologies at individual and population levels for treatment remains ongoing [107].

Additionally, the current availability of safe and effective DAA therapy for HCV plays a central role in combating the virus. These therapies have the potential to halt viral transmission from an infected individual to a susceptible subject. However, their high cost and limited access in LMICs present significant challenges.

In 2015, the WHO adopted an ambitious, albeit perhaps overly optimistic, program to eliminate HCV as a public health threat by 2030 [108]. The targets of this program include a 90% reduction in incident cases, a 65% reduction in mortality, and ensuring that 80% of treatment-eligible subjects have access to care. Achieving these targets requires universal screening, diagnosis, linkage to care, and access to affordable DAA regimens. However, these goals are particularly challenging to reach in LMICs, as well as in rural and forested areas in South America, Africa, and Southeast Asia. Concerns about the feasibility of this project stem from issues such as the cost and supply of these drugs and the complexities of the logistic chain in these regions.

An effective vaccine may represent the most promising pathway to achieve elimination of this infectious disease. Unfortunately, an effective vaccine against HCV has not yet been developed. A recent phase 2 randomized trial of an HCV vaccine yielded unsuccessful results: while the vaccine reduced HCV RNA levels, it did not prevent chronic HCV infection [109]. Much work remains to be done in this area. In the absence of an effective vaccine, the WHO project to eliminate HCV remains just a dream.

## 4. Conclusions

HCV infection poses a significant global health challenge. While high-income countries have made notable progress in reducing certain modes of transmission, such as ensuring the safety of blood and blood products and, to a lesser extent, minimizing iatrogenic exposure, low- and middle-income countries (LMICs) continue to grapple with issues like unscreened blood transfusions and non-sterile injections, highlighting the stark inequality between these two settings. Socio-economic factors, healthcare infrastructure limitations, and cultural barriers influence HCV transmission and control efforts, particularly in more resource-limited settings. Intravenous drug use remains a significant challenge, with its prevalence also on the rise in LMICs.

As presented above, the evolving modes of transmission over time reported in Table 5 explain the current birth cohort prevalence of individuals infected with HCV, with an emerging shift toward younger generations. The impact of the evolving drug use pattern will represent a worldwide challenge in the years to come, requiring major attention. In this context, the introduction of new digital and non-digital technologies, the pathways of evolving treatment, and the potential introduction of a vaccine would be crucial. While curative DAA therapy has been identified as a means to halt HCV transmission, the lack of an effective vaccine remains the primary barrier to achieving HCV elimination on a global scale.

## Figures and Tables

**Figure 1 viruses-16-01115-f001:**
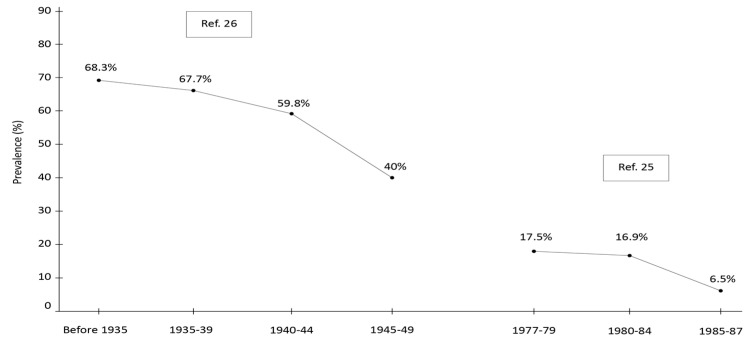
Anti-HCV positivity prevalence in Cameroon. Adapted from References [21,29].

**Figure 2 viruses-16-01115-f002:**
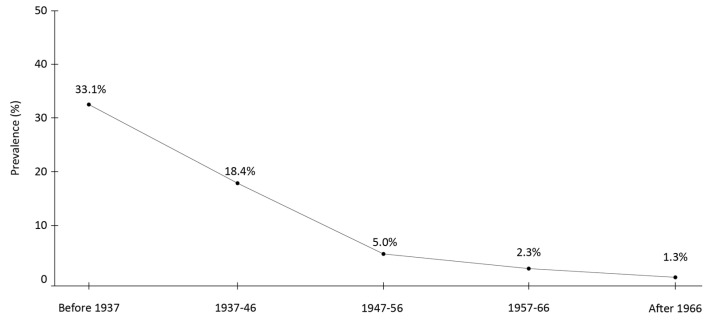
Rate of anti-HCV positivity in Italy according to birth cohort. Adapted from [24].

**Figure 3 viruses-16-01115-f003:**
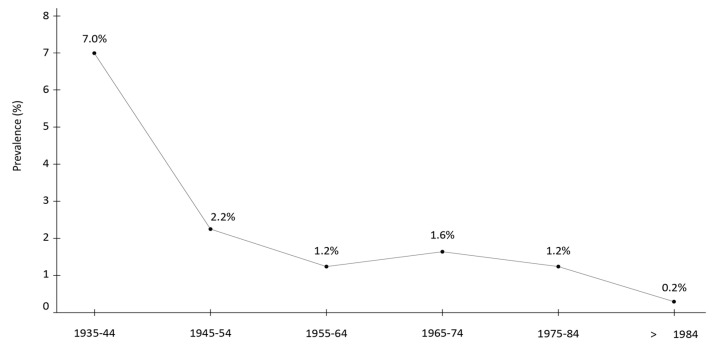
Rate of anti-HCV positivity in Italy according to birth cohort, as assessed in five metropolitan areas.

**Figure 4 viruses-16-01115-f004:**
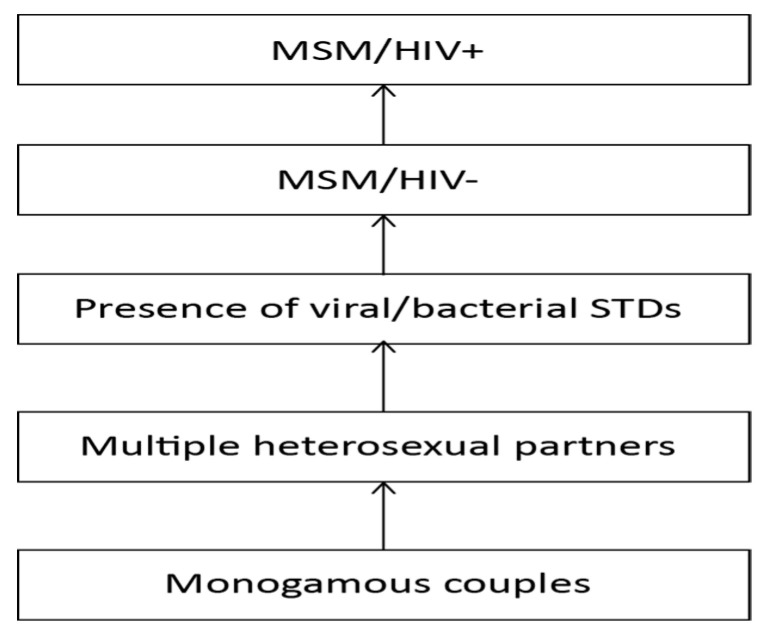
Gradient of increasing risk of sexual HCV transmission according to the setting of intercourse.

**Table 1 viruses-16-01115-t001:** Prevalence (%) of HCV infection among the general population in 2019, by WHO region. Adapted from Reference [6].

Africa	0.8
Americas	0.5
South-East Asia	0.5
European Region	1.3
Eastern Mediterranean	1.6
Western Pacific	0.5

**Table 2 viruses-16-01115-t002:** Countries with widespread of HCV infection during the 20th century.

Country	Estimated Years of Start	Overall Prevalence	Mode of Transmission	Reference N.
Egypt	1930s	24.3%	I.V. Treatment for Schistosomi	[19,20]
Cameroon	1920s	40–68%	I.V. Treatment for Malaria	[21]
Central Africa Republic	1940–50s	10.5%	Treatment for Thrypanosomiasis	[22]
Japan	1960s	32.4%	Parenteral folk remedies	[23]
Italy	1950–60s	12.6%	Parenteral therapy	[24]

**Table 3 viruses-16-01115-t003:** Prevalence of anti-HCV in an isolated area in Japan 1995. Adapted from Reference [23].

Age Group (Years)	Anti-HCV Prevalence
5–15	0
16–40	12.5%
>40	44.9%
Overall	32.4%

**Table 4 viruses-16-01115-t004:** Prevalence (%) of anti-HCV in a southern Italian town, 14 years apart.

Age Group (Years)	1996 (Ref. [24])	2010 (Ref. [35])
<30	1.3	0.4
30–39	2.3	0.7
40–49	5.0	1.5
50–59	18.4	1.1
≥60	33.1	12.0
Overall	12.6	5.7

**Table 5 viruses-16-01115-t005:** Modes of HCV transmission over time, by income of countries. Number of * in the figure indicates the relative impact of those modes of HCV transmission.

Modalities	High Income		Low/Medium Income	
	Past Decades	Currently	Past Decades	Currently
Blood Transfusion	*****	Near 0	*****	****
PWID	****	*****	**	***
Unsafe medical equipment	****	**	*****	****
Risky sexual behavior	***	***	***	***
Vertical transmission	*	**	*	**
Body grooming and modification	**	***	***	***

**Table 6 viruses-16-01115-t006:** Residual risks of parenterally transmitted viruses in pre- and during NAT eras, by number of donations.

Virus	Pre-NAT (1999–2001) Ref. [52]	During NAT (2009–2015) Refs. [53,54]
HCV	16.7/1 million donations	1/12,979,949 donations
HIV	1.9/1 million donations	1/1,917,250 donations
HBV	69.2/1 million donations	1/2,566,844 donations

**Table 7 viruses-16-01115-t007:** Adjusted odds ratios (O.R.) for the association of HCV infection with tattooing.

Country	O.R. (CI: 95%)	Patients	Year	Reference
Taiwan	5.9 (1.6–22.0)	Chronic HCV	1992	[90]
Norway	5.4 (1.7–9.2)	Chronic HCV	1993	[91]
New Mexico	5.9 (1.1–30.7)	Chronic HCV	1999	[92]
Texas	6.5 (2.9–14.7)	Chronic HCV	2001	[93]
Italy	5.6 (2.8–11.0)	Acute HCV	2004	[94]

## Data Availability

Not applicable.

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
