# Peer review of "Prevalence and Modes of Transmission of Hepatitis C Virus Infection: A Historical Worldwide Review"

_viruses, 2024, doi:10.3390/v16071115_

Round 1
Reviewer 1 Report (New Reviewer)
Comments and Suggestions for Authors
The article entitled “Prevalence and modes of transmission of hepatitis c virus infection: a historical worldwide review” is intended to describe the transmission of hepatitis c virus (HCV) under different perspectives.
Comments:
1. In this review. The ABSTRACT represents some compilation of extremely general nature and it seems that the article has been compiled as one of the several articles of this nature. A snap shot of HCV infection and its incidence and prevalence has been provided with a generalized tone about its containment in global basis. I assume that the ABSTRACT must be improved so that it gets it proper instinct.
2. The authors should be credited for compiling the second part of the INTRODUCTION. They have provided a discussion of historical nature of HCV infection. Although Table 2. Has provided a comprehensive note regarding HCV epidemic in 20th centuries. The authors should clarify some points and these should be discussed with evidences. The first, if the use of word “epidemic” is valid here. The next, HCV has been discovered in 1989. Is it rationale to use the term “HCV Epidemic” for the illnesses in early 2oth century. Even with these limitations, I would provide high marks to the authors for such a beautiful compilation of history of HCV infection.
3. The authors have discussed regarding several risk factors related to HCV transmission. However, one such route has been omitted by the authors. In Pakistan, HCV is mainly transmitted via shaving with the same blade.
4. In the Abstract, there are some descriptions of HCV elimination by traditional DAA and proper use of these drugs may be somehow ambitious for the entire world, especially the n low- and middle-income countries (LMICs). Under these situations, the present status of “Elimination of Hepatitis by 20230” should be discussed in the light of HCV elimination. As the declaration was taken in 2015 and 9 years have passed, we have only 6 years in hand. A proper assessment of these factors may aid the ultimate elimination.
Author Response
Reviewer 1
The article entitled “Prevalence and modes of transmission of hepatitis c virus infection: a historical worldwide review” is intended to describe the transmission of hepatitis c virus (HCV) under different perspectives.
Comments:
- In this review. The ABSTRACT represents some compilation of extremely general nature and it seems that the article has been compiled as one of the several articles of this nature. A snap shot of HCV infection and its incidence and prevalence has been provided with a generalized tone about its containment in global basis. I assume that the ABSTRACT must be improved so that it gets it proper instinct.
Response
Thank you for your constructive feedback. We have revised the Abstract to provide a more focused and informative overview
- The authors should be credited for compiling the second part of the INTRODUCTION. They have provided a discussion of historical nature of HCV infection. Although Table 2. Has provided a comprehensive note regarding HCV epidemic in 20th The authors should clarify some points and these should be discussed with evidences. The first, if the use of word “epidemic” is valid here. The next, HCV has been discovered in 1989. Is it rationale to use the term “HCV Epidemic” for the illnesses in early 2oth century. Even with these limitations, I would provide high marks to the authors for such a beautiful compilation of history of HCV infection.
Response
We appreciate your acknowledgment of the historical perspective provided in the Introduction. We have revised the terminology from "epidemic" to "widespread" to accurately reflect the historical context of HCV infection throughout the 20th century.
- The authors have discussed regarding several risk factors related to HCV transmission. However, one such route has been omitted by the authors. In Pakistan, HCV is mainly transmitted via shaving with the same blade.
Response
Thank you for highlighting the importance of including diverse transmission routes. We have incorporated findings from Pakistan, specifically addressing HCV transmission via shared shaving blades, and have cited additional relevant literature (Reference [97]).
- In the Abstract, there are some descriptions of HCV elimination by traditional DAA and proper use of these drugs may be somehow ambitious for the entire world, especially the n low- and middle-income countries (LMICs). Under these situations, the present status of “Elimination of Hepatitis by 20230” should be discussed in the light of HCV elimination. As the declaration was taken in 2015 and 9 years have passed, we have only 6 years in hand. A proper assessment of these factors may aid the ultimate elimination.
Response
We have discussed the limitations of direct-acting antiviral (DAA) treatments for achieving global HCV elimination, especially in low- and middle-income countries (LMICs). This discussion provides a critical assessment of the feasibility of reaching the goal of hepatitis elimination by 2030, considering the current status and challenges faced.
Reviewer 2 Report (New Reviewer)
Comments and Suggestions for Authors
The authors provide an interesting review on HCV epidemiology, with particular description on the historical evolution of high prevalences of HCV associated with particular iatrogenic practices.
One minor concern: it would be interesting to add additional explanation (in addition to the reduction of iatrogenic transmission) on how Africa comes now to exhibit such a low prevalence (0.8%) (Table 1). Is the HCV prevalence accurately estimated in each African country?
Author Response
Reviewer 2
The authors provide an interesting review on HCV epidemiology, with particular description on the historical evolution of high prevalences of HCV associated with particular iatrogenic practices.
One minor concern: it would be interesting to add additional explanation (in addition to the reduction of iatrogenic transmission) on how Africa comes now to exhibit such a low prevalence (0.8%) (Table 1). Is the HCV prevalence accurately estimated in each African country?
Response
We appreciate your insightful comment regarding the low prevalence of HCV in Africa (0.8%) as indicated in Table 1. In response, we have expanded our discussion to include additional explanations beyond the reduction of iatrogenic transmission. This includes considerations on the accuracy of HCV prevalence estimates across African countries.
These revisions aim to provide a more comprehensive explanation regarding the factors contributing to the current low prevalence of HCV in Africa, addressing your valuable input.
Reviewer 3 Report (New Reviewer)
Comments and Suggestions for Authors
I read with interest the review entitled “Prevalence and modes of transmission of hepatitis C virus infection: a historical worldwide review”. The authors gave a historical perspective on the prevalence and modes of transmission of HCV in low and middle income, as well as in western countries. The paper is well written, and data are logically presented. I do not have any major complaints.
Minor changes:
- English editing is required
- Table 7 – add in Paragraph 3. (Modes of transmission), not in conclusion
- Paragraph 3.1 Perinatal transmission – consider adding the Table or Figure with screening recommendations and practices in pregnant women for anti-HCV. Even in high-income countries this is not routine practice.
- 3.6. Beauty treatment transmission. Consider another title. This might be misleading since you are talking about tattooing and piercing, while some readers could expect data on HCV transmission during “beauty treatment” as visiting nails saloons, cosmetic treatments…
Comments on the Quality of English LanguageModerate editing required.
Author Response
Reviewer 3
I read with interest the review entitled “Prevalence and modes of transmission of hepatitis C virus infection: a historical worldwide review”. The authors gave a historical perspective on the prevalence and modes of transmission of HCV in low and middle income, as well as in western countries. The paper is well written, and data are logically presented. I do not have any major complaints.
Minor changes:
- English editing is required
- Table 7 – add in Paragraph 3. (Modes of transmission), not in conclusion
- Paragraph 3.1 Perinatal transmission – consider adding the Table or Figure with screening recommendations and practices in pregnant women for anti-HCV. Even in high-income countries this is not routine practice.
- 3.6. Beauty treatment transmission. Consider another title. This might be misleading since you are talking about tattooing and piercing, while some readers could expect data on HCV transmission during “beauty treatment” as visiting nails saloons, cosmetic treatments…
Response
Dear Reviewer,
Thank you very much for your thorough assessment of our manuscript. Your constructive comments have improved the revised version of the manuscript. Please find our responses below:
- English Editing: The manuscript has undergone thorough English editing to ensure clarity and coherence.
-Table 7 Placement: Table 7 has been relocated to Paragraph 3 (Modes of transmission) as suggested.
- Recommendations from international societies are presented in the text with cited references, in particular: “In 2019, AASLD–IDSA guidelines, subsequently updated in 2023, recommended routine HCV testing for pregnant women, proving cost-effective if HCV prevalence among them is 0.03% or higher [82]. This aligns with the 2020 universal screening recommendations by the US Preventive Services Task Force [83]. Currently, obstetrical societies and general screening recommendations, including WHO, mostly rely on targeted risk or birth cohort-based strategies to detect HCV in pregnant women, but screening practices are expected to change as countries work toward HCV elimination goals. Unfortunately, the implementation of these testing strategies is challenging in LMICs. Furthermore, risk factors for HCV infection, such as female genital mutilation, prevalent in various African regions, may not be adequately accounted for in existing methodologies [84]”.
Including a comprehensive table or figure encompassing all global recommendations may not be practical in the scope of this review on the worldwide mode of transmission of HCV. Nevertheless, we have augmented the discussion with additional data and reference on this topic, which the reviewer can now find in the paragraph. These challenges are extensively discussed in the text, providing further insight into the complexities surrounding HCV screening practices worldwide, in particular: "In the document “Global Hepatitis Report 2024: Action for Access in Low- and Middle-Income Countries,” it is noted that most reporting countries have adopted and are implementing a national hepatitis C testing strategy or policy in line with WHO guidelines. While nearly all (90%) provide targeted testing for individuals from the most severely affected populations, only six countries focus their efforts on testing pregnant women. Consequently, implementing these specific testing strategies remains challenging in low- and middle-income countries [84]"
- Title Modification for Paragraph 3.6: The title "3.6. Beauty treatment transmission" has been revised to "3.6. Body grooming and modification" to more accurately reflect the content discussing tattooing, piercing, and related practices. This adjustment aims to avoid misleading expectations among readers.
This manuscript is a resubmission of an earlier submission. The following is a list of the peer review reports and author responses from that submission.
Round 1
Reviewer 1 Report
Comments and Suggestions for Authors
The authors provide a comprehensive overview of HCV transmission which includes historical accounts of HCV transmission prior to the current epidemics that surround injection drug use and unsafe injection practices and blood products, particularly in LMIC. I have no significant major issues with the manuscript, but have several minor comments noted below:
Overall comments.
· Please change wording of “intravenous drug use” to the updated wording of “injection drug use” in abstract, introduction, and other places in the paper where this terminology is used. Also change “injective drug use” to injection drug use (page 6 first paragraph).
· In the introduction it would be useful if the authors could give a little more context surrounding drug injection as it relates to HCV transmission; while blood transfusions are clearly a vector for transmission, among many locations, injection drug use continues to drive infection, particularly in high income locations and areas where blood products are routinely tested for HCV.
· In the “prevalence” section the authors should note that many locations are not able to reliably report HCV infection due to cost and logistical constraints surrounding being able to have a surveillance system in place to detect and document new and existing cases of HCV.
· Section 2.3: prevalence by sex: in most locations, injection drug use is more highly prevalent in males compared to females; this is likely another reason that there is a high ratio of male to female infections and is worth mentioning in thi section.
· On page 8, I am not sure what this sentence is trying to say “Consequently, the risk of transmission through contaminated needles and syringes after needle exchange programs may persist for a longer period with HCV”. Also, it would be helpful to highlight that the amount of HCV virus that is needed to transmit the virus through contaminated equipment is much smaller than for HIV, which also contributes to increased efficiency of transmission among persons who inject drugs.
· In section 3.6, it would be helpful to expand this discussion to highlight tattooing risk factors for HCV transmission in prison settings; it is well known that prison inmates receive tattoos while incarcerated as a sign of gang affiliation and for other personal reasons, and they are often done using non-sterilized or rudimentary instruments, increasing the risk of blood borne virus transmission in a setting where HCV is already prevalent (see https://harmreductionjournal.biomedcentral.com/articles/10.1186/s12954-015-0045-2)
Comments on the Quality of English Languageplease change some of the terms for injection drug use as noted above to the chosen accepted terms.
Author Response
Firstly, thanks for the careful assessment and the useful suggestions.
- Wording has been changed in all places.
- In the introduction, the relevance of injection drug use has been highlighted.
- In the prevalence section a new sentence has been added.
- Section 2.3: the suggested sentence has been added.
- On page 8, some typewritten errors have generated confusion; the sentence has been reworded. The sentence suggested by reviewer has been added.
- In section 3.6, the role of tattooing in prison setting has been highlighted. A new reference has been added.
Reviewer 2 Report
Comments and Suggestions for Authors
The review titled: “Prevalence and modes of transmission of hepatitis c virus infection: a historical worldwide review” by Stroffolini & Stroffolini attempts to focus on the global pattern of HCV prevalence and transmission modes over time, to highlight their importance in defining preventive and curative measures, as well as identify potential future vaccination strategies. The text provides a comprehensive overview of the modes of transmission of HCV and the efforts to combat its spread. Each mode is discussed in detail, highlighting its significance, associated risks, and potential preventive measures. It covers various aspects, including historical perspectives on transmission, current trends, challenges, and interventions, discussing not only the epidemiological aspects but also the underlying factors contributing to transmission, such as social, economic, and healthcare system-related factors. It also examines the effectiveness of interventions and identifies gaps and challenges in addressing HCV transmission. The information presented in the text appears to be accurate and well-supported. It cites relevant studies, data, and statistics to substantiate the claims made regarding the prevalence of HCV transmission, effectiveness of interventions, and challenges faced in different settings.
Overall, the review is well-written, offering a comprehensive and well-researched examination of the modes of transmission of HCV and the efforts to eliminate its spread. It provides valuable insights for healthcare professionals, policymakers, and researchers working in the field of public health and infectious diseases. It maintains a balanced perspective by acknowledging both the progress made in reducing HCV transmission, particularly in high-income countries, and the persistent challenges faced, especially in low- and middle-income countries, and recognizes the disparities in healthcare infrastructure and access that contribute to unequal outcomes in different regions.
However, a few points should be addressed for improvement:
1. Clarification on Preventive Measures: While the text discusses preventive measures briefly, it could benefit from elaborating more on specific interventions and strategies aimed at reducing HCV transmission. Providing additional information on the implementation and effectiveness of preventive measures, such as harm reduction programs for intravenous drug users or the promotion of safe medical practices, would enhance the comprehensiveness of the discussion.
2. Global Perspective: While the text touches upon the disparities in HCV transmission and prevention efforts between high-income countries and LMICs, it could delve deeper into the unique challenges faced by different regions and populations. Exploring the socio-economic factors, healthcare infrastructure limitations, and cultural barriers that influence HCV transmission and control efforts in diverse settings would provide a more comprehensive understanding of the global impact of the disease.
3. Future Directions: It would be beneficial to include a section or discussion on emerging trends in HCV transmission and potential future directions for research and intervention. This could involve addressing issues such as the impact of evolving drug use patterns, the introduction of new treatment modalities, the role of digital health technologies in healthcare delivery, and an expanded discussion on achieving the WHO's goal of eliminating HCV as a public health threat by 2030.
Author Response
Thanks for having appreciated our efforts
- In the section 3.7, page 10, preventive measures were already mentioned.
- New sentences have been added in Conclusions.
- Similarly, new sentences have been added in Conclusions.
Reviewer 3 Report
Comments and Suggestions for Authors
In this manuscript, the authors described HCV’s prevalence and transmission mode. The manuscript concisely described the historical story of HCV using previous publications. The manuscript could be helpful to beginner scientists who would study HCV; however, the manuscript contains outdated data and well-known information about HCV. This reviewer want to point out several outdated data pieces that are outdated and probably used in many reviews. The manuscript should use more recent and updated information. Otherwise, the review is of no use to the scientist.
1. Table 2 contains 1920~1960s information.
2. Figure 1 contains the information before 1987.
3. Figure 2 contains the information before 1966.
4. Table 4 contains the information before 2010.
5. Figure 3 contains the information before 1984.
Author Response
We thank the reviewer for his assessment. Nonetheless, we disagree with the reviewer comments that the manuscript could be useful only for beginning scientists and that it contains just outdated data.
As reported in the Title , this is an historical review. Report of epidemics is relevant to substantiate the over-time burden and modes of transmission of HCV infection in different areas. The paper covers historical perspectives, current trends, and challenges. It may be useful even for skilled scientists, healthcare professional, and researchers working in the field of infectious diseases and public health.
Several references are updated; you can easily check that.
The reviewer cites some Tables and Figures as proof of the use of outdated findings. These data used for Tables and Figures come from studies mostly published after the year 2010; data presentation has been made plotting HCV prevalence by birth-cohort. It means that the HCV prevalence of the enrolled population (including oldest generations) was plotted by the years of birth, allowing the evidence for the HCV cohort effect, i.e. decreasing risk of exposure to the infection along generations.
Round 2
Reviewer 2 Report
Comments and Suggestions for Authors
Nothing that I asked was addressed.
1. While the text discusses preventive measures briefly, it could benefit from elaborating MORE. You did not address this point at all.
2 and 3. I fail to see how any of the sentences added addressed any of the questions I posed.
Author Response
Dear Reviewer,
Thank you for your assessment of our manuscript.
We respectfully disagree with the assertion that none of the reviewer's concerns were addressed. The review focuses on "Prevalence and modes of transmission of hepatitis C virus infection: a historical worldwide review."
In the initial report, the reviewer acknowledges that the review provides a comprehensive overview of HCV transmission modes, preventive measures, challenges, and interventions, supported by relevant literature.
Following the first round of revisions, incorporating the reviewer's suggestions, we have expanded the text and provided new references. We integrated some of Reviewer 2's comments to enhance:
1) clarification on preventive measures
2) exploration of global perspectives
3) touching on future directions
While we may not have fully addressed all suggestions, we have broadened the discussion without deviating from the manuscript's focus.
To meet the new requirements, we have updated the text with additional information, bringing the word count to over 5000 words, approaching the 6000-word limit. Specifically, we have taken a broader approach to clarify preventive measures, explore global perspectives, and discuss future directions.
In the "injection drug use" section, we have incorporated recent data from a systematic review and meta-analysis, highlighting interventions aimed at overcoming barriers to testing, linkage to care, and treatment initiation worldwide.
For the "sexual transmission" section, we have included more data relevant to the MSM or HIV-positive population, focusing on addressing risk factors and the role of HCV spontaneous clearance.
Furthermore, in the "perinatal transmission" section, we have added more information regarding universal screening and testing strategies in various settings, encompassing both high and low-middle-income countries.
Lastly, we have updated the "Attempts to eliminate HCV infection" section with fresh data, discussing the potential contribution of digital health technologies (machine learning, artificial intelligence) in healthcare delivery.
Thank you once again for your continued guidance, which has enriched the manuscript.
Reviewer 3 Report
Comments and Suggestions for Authors
Although the authors tried to add more updated references to the review manuscript, they added many references in just one sentence. This reviewer asked the authors to add the recent references in every section to make the manuscript updated. Therefore, the responses to the reviewer were not sufficiently satisfactory and the authors should update every chapter with new recent references.
Author Response
Dear reviewer
thank you for assessing this manuscript.
As suggested, we now included further updated references in the text in different chapters.